# Two New Species of *Encotyllabe* (Monogenea: Capsalidae) from Brazil: Morphological and Molecular Evidence

Naraiana Taborda [1], Fabiola A. Sepulveda [2], Jose L. Luque [3], Rubén Escribano [4] and Marcelo E. Oliva [2,4,*,†]

1    Programa de Pós-Graduação em Ciências Veterinárias, Universidade Federal Rural do Rio de Janeiro, Seropédica 23851-970, Brazil; naraiana.t@hotmail.com
2    Instituto de Ciencias Naturales Alexander von Humboldt, Universidad de Antofagasta, Antofagasta 1240000, Chile; fabiolasepu@gmail.com
3    Departamento de Parasitologia Animal, Universidade Federal Rural do Rio de Janeiro, Seropédica 23851-970, Brazil; luqueufrrj@gmail.com
4    Instituto Milenio de Oceanografía, Universidad de Concepción, Concepción 4030136, Chile; ruben.escribano@imo-chile.cl
*    Correspondence: marcelo.oliva@uantof.cl; Tel.: +56-55-6237404
†    Zoobank: lsid:zoobank.org:act:0A209ECA-3192-4AA5-ABD3-678D0FFFDF2A and lsid:zoobank.org:pub:2422AA43-56A7-4B88-A9B3-DFF5BAF2FDEB.

**Abstract:** Currently, 24 species of *Encotyllabe* Diesing, 1850 (Monogenea: Capsalidae), are recorded as parasites on teleost fishes, but the validity of many species has been questioned due to deficient or incomplete descriptions. Almost all species in the genus have been described from one host species or closely related host species, suggesting host specificity, but some species, specifically *Encotyllabe spari* Yamaguti, 1934, have been reported from at least 19 species belonging to nine families in two orders (Perciformes and Scorpaeniformes) from Japan, Arabian Gulf and Brazil. Concerning Brazilian records of *Encotyllabe spari* and *Encotyllabe* cf. *spari*, seven species belonging to four families and two orders have been reported as hosts for this species. The aim of this study was to describe two new species of *Encotyllabe* from Brazil, previously considered as *E. spari*. Morphological and morphometric (multivariate analysis of proportional measurements standardized by total length) and molecular analysis (LSU rRNA and cox1 gene) were performed in order to identify the collected monogeneans. The description of two new species, namely *Encotyllabe bifurcatum* n. sp. and *Encotyllabe parvum* n. sp., parasitizing *Pagrus pagrus* and *Orthopristis ruber*, respectively, is the result of the three approaches. The main morphological differences from the most related species include a combination of body size, shape of the male copulatory organ, size and position of the testes. Our results suggest host specificity for members of *Encotyllabe* and specimens previously recorded as *E. spari*, other than those from the original description, must be revisited.

**Keywords:** Capsalidae; *Encotyllabe parvum* n. sp.; *Encotyllabe bifurcatum* n. sp.; Southwestern Atlantic; host specificity; LSU rRNA gene; cox1 gene

## 1. Introduction

To date, 24 species of *Encotyllabe* Diesing, 1850 (Monogenea: Capsalidae), are listed in WoRMS [1]; *Encotyllabe callaoensis* was originally described in an unpublished thesis [2] and never published according to the rules of the International Code for Zoological Nomenclature; this species is not considered herein as valid as previously suggested [3]. There is consensus that some species of the genus should be considered species inquirenda, mainly by identifications based on one or two specimens without taking into account morphological intraspecific variations, in addition to the variation in the measurements and distortion in the position of organs, resulting from fixation and flattening the worms [3–5]. The first four species described in the genus, *E. nordmanni* Diesing, 1850, from *Brama rail* (Bonnaterre) in the Mediterranean Sea, *E. pagelli* Van Beneden and Hesse, 1863, from *Pagellus centrodontus*

(Brünnich) in Brest, France, *E. paronae* Monticelli, 1907, from *Crenilabrus pavo* (Linnaeus), in Geneves, and *E. vallei* Monticelli, 1907, from *Chrysophrys aurata* (Linnaeus) in Trieste, were so briefly described or illustrated that their similarities to, and differences from, other species are difficult to assess with absence of morphometric data, and thus they should be considered *species inquirenda*. For instance, *E. nordmanni* was described as "longit. Corp 11/2, latit 1/2, longit. pedic. Acet 1/2." [6]. The same applies for *E. masu* Ishii and Sawada, 1938, *E. monticelli* Perez Vigueras, 1940, *E. pricei* Koratha, 1955, and *E. punctatai* Gupta and Krishna, 1980, which have been considered *species inquirenda* because they were poorly described and are based on only one or two specimens [3,4]. The status of those species must be confirmed following redescriptions based on specimens from the type host and locality due to the frequent host specificity of capsalids [7] as well as the appreciable geographic variation observed in some species of monogeneans [8,9].

A particular situation occurs with *E. lintoni* Monticelli, 1909. This species was described based on one anomalous specimen bearing one testis and a small scar-like mass of tissue in place of the other testis, from a specimen from *Calamus calamus* (Valenciennes) (Sparidae) caught in Bermuda and redescribed [10] based on the same specimen of the original description as well as specimens from *Pagrus pagrus* (Linnaeus) (Sparidae) in Brazil without vouchers deposited in a curated collection [11].

Here, 14 species of *Encotyllabe* are considered as valid: *E. antofagastensis* Sepúlveda, González and Oliva, 2014, *E. caballeroi* Velasquez, 1977, *E. caranxi* Lebedev, 1967, *E. cheilodactyli* Sepúlveda, González and Oliva, 2014, *E. chironemi* Robinson, 1961, *E. embiotocae* Noble, 1966, *E. fotedari* Gupta and Krishna, 1980, *E. kuwaitensis* Khalil and Abdul-Salam, 1988, *E. lintoni* Monticelli, 1909, *E. lutjani* Tripathi, 1959, *E. pagrosomi* MacCallum, 1917, *E. souzalimae* Carvalho and Luque, 2012, *E. spari* Yamaguti, 1934 and *E. xiamenensis* Li, Yan and Wang, 2004. Recently [12], molecular data (cox1 and 28S gene) were given for two unnamed, nor described, species of *Encotyllabe*, called *Encotyllabe* sp.1 and *Encotyllabe* sp. 2.

The most reliable taxonomic criteria for mature specimens of *Encotyllabe* are body shape, relative size of several organs, shape and position of testes, distance between the center of the ovary and center of the testis, male copulatory organ (MCO) shape, extension of the vitellaria and length of peduncle in relation to the body [3,4,13,14]. However, multivariate analysis of proportional measurements standardized by total length as well as molecular studies will strongly improve the taxonomy of *Encotyllabe* [3] and also other monogeneans [15].

The following species of *Encotyllabe* have been recorded from marine fishes in Brazil: *Encotyllabe spari* in *Haemulon sciurus*, *Orthopristis ruber*, *Anisotremus surinamensis*, *Conodon nobilis* (Haemulidae), *Pagrus pagrus* (Sparidae), *Menticirrhus americanus*, *Micropogonias furnieri* (Sciaenidae) and *Dactylopterus volitans* (Dactylopteridae), *Encotyllabe* cf. *spari* in *O. ruber*, *Encotyllabe lintoni* in *P. pagrus*, and *Encotyllabe souzalimae* in *Trichiurus lepturus* (Trichiuridae) and *Thyrsitops lepidoides* (Gempylidae) [4,11,16–25].

Since *Encotyllabe spari* have been reported from Brazilian fishes belonging to seven species in four families and two orders (Perciformes and Scorpaeniformes), we searched for this species in *P. pagrus* and *O. ruber*, the most commonly reported hosts in Brazil, in order to confirm their identity. Fresh material allowed us to perform molecular as well as morphological and morphometric analyses. Our results indicated that specimens previously reported as *E. spari* in *P. pagrus* and *O. ruber* belong to two new species. These are described and differentiated below.

## 2. Materials and Methods

### 2.1. Sample Collection and Processing

During 2016, 26 specimens of *Pagrus pagrus* and 9 specimens of *Orthopristis ruber* were obtained from local fishermen off Cabo Frio, Rio de Janeiro, Brazil. In the laboratory, fish were dissected, and gills and pharyngeal plates were removed and examined under a stereomicroscope for monogeneans. Thirty-four specimens of two species of *Encotyllabe*

obtained from *P. pagrus* (n = 16) and *O. ruber* (n = 18) were selected for morphological and molecular analyses.

Some worms were fixed in 4% neutral buffered formaldehyde and then transferred and stored in 70% ethanol for further morphological studies (light microscopy). Selected specimens from each of the two host species were transferred to 96% ethanol for DNA analysis. Population descriptors, prevalence and mean intensity [26] were recorded.

### 2.2. Morphological and Statistical Analysis

Fixed specimens were stained with carmine, cleared with clove oil (Sigma-Aldrich, Taufkirchen, Germany) and mounted in Canada balsam. Drawings were made with the aid of an Olympus BX53 microscope (Olympus Corporation, Tokyo, Japan) equipped with a drawing tube. Measurements are given in micrometers unless otherwise stated, with a range followed by the mean and number of measurements (n) in parentheses (see Supplementary Material Table S1 for comparative measurements). Type-material was submitted to the Helminthological Collection Instituto Oswaldo Cruz (CHIOC)—Rio de Janeiro—Brazil. For comparative purposes, vouchers of *Encotyllabe spari* from Brazil deposited in the Helminthological Collection of the Oswaldo Cruz Institute (CHIOC) from *P. pagrus* (CHIOC 34531), *Haemulon sciurus* (Shaw) (CHIOC 32019, 32064, 32068), *Anisotremus virginicus* (Linnaeus) (CHIOC 37947), and *Conodon nobilis* (Linnaeus) (CHIOC 37948) were studied.

### 2.3. Morphometric Analysis

Multivariate morphometric analyses were performed on 16 specimens from *O. ruber* and nine from *P. pagrus*. Principal component analysis (PCA) was used for proportional morphometric measurements, and because of the expected correlation between the size of different organs and body length (BL, excluding peduncle), we used proportion rather than raw measurement [3,15]. The proportion measurements were as follows: (1) maximum body width/BL, (2) length of the peduncle/BL, (3) diameter of haptor/BL, (4) diameter of the anterior attachment organs/BL, (5) pharynx width/BL, (6) length of the ovary/BL, (7) ovary width/BL, (8) length of the testes/BL, (9) width of the testes/BL, (10) length of the large anchors/BL, (11) length of the small anchors/BL, and (12) length of the marginal hooks/BL. PCA was performed using Statistic 7.0 software (Statsoft Inc., Tulsa, OK, USA).

### 2.4. Molecular Data and Phylogenetic Analyses

DNA was isolated from each individual following a modified protocol [27] involving treatment with sodium dodecyl sulphate, digestion with proteinase K, NaCl protein precipitation and subsequent ethanol precipitation. DNA was eluted in nuclease-free water and quantified in a Biospec-nano spectrophotometer. For molecular analyses, the nuclear LSU rDNA and mitochondrial gene cytochrome c oxidase 1 (*cox*1) were used. The LSU rRNA gene was amplified with the forward primer C1 (5′-ACCCGCTGAATTTAAGCAT-3′) and reverse primer D2 (5′-TGGTCCGTGTTTCAAGAC-3′) [28]. The *cox*1 gene was amplified with the forward primer ASmit1 (5′-TTTTTTGGGCATCCTGAGGTTTAT-3′) and reverse primer ASmit2 (5′-TAAAGAAAGAACATAATGAAAATG-3′) [29]. Each PCR had a final volume of 35 μL, including five standard units of GoTaq DNA polymerase (Proma, Madison, NJ, USA), 7 μL 5× PCR buffer, 5.6 μL $MgCl_2$ (25 mM), 2.1 μL BSA (10 mg/mL), 0.7 μL of deoxynucleotide triphosphate (dNTP) (10 mM), 10 pM of each primer, 3 μL template DNA and sufficient nuclease-free $H_2O$ to reach a total volume of 35 μL. A Boeco Ecogermany M-240R Thermal Cycler (Boeckel, Hamburg, Germany) was used to carry out PCR. Amplification for each molecular marker follows [28,30] for the LSU rRNA and *cox*1 gene, respectively. Both DNA strands were directly sequenced (Macrogen, Seoul, Korea; http://www.macrogen.com, accessed on 20 May 2022). Sequences were edited and contigs assembled using ProSeq v 2.91 beta [31].

For each gene, a database was constructed in FASTA format (new generate sequences + sequences of *Encotyllabe* spp. from GenBank, see Table 1) and aligned with Clustal X [32].

A visual inspection was performed with ProSeq v.2.91 [31] to edit the length of the final data set. Sequences of *Encotyllabe* spp. were analyzed separately according to each studied gene (Table 1), and host and those sequences that presented polymorphism between them were selected for phylogenetic analysis.

Phylogenetic reconstructions were conducted independently for each gene fragment. Bayesian inference (BI) was analyzed with MrBayes [33] and maximum likelihood (ML) analysis with W-IQ-TREE [34].

Each set of aligned sequences was analyzed with the software jModelTest 0.1.1 [35], which compares different models of DNA substitution in a hierarchical hypothesis-testing framework to select a base substitution model that best fits the data for each gene. The optimal models found by jModelTest, selected with the corrected Akaike information criterion, was GTR + G for the LSU rRNA gene and TIM2 + G + I for the *cox*1.

Bayesian inference analyses were executed with the following parameters: nst = 6, rates = gamma for LSU rRNA and nst = 6, rates = invgamma according to the evolutionary model determined by jModeltest 0.1.1 for each gene. The analysis was performed for 5,000,000 generations. Analyses included two runs of four chains and sampling every 1000 generations. Support for nodes in the BI tree topology was obtained by posterior probability burn-in of the initial 25% of samples. Visual inspection of log likelihood scores against generation time was performed in TRACER v.1.7 [36]. Statistical support for ML analyses was performed with 1000 bootstraps. The trees were visualized and edited in FigTree v.1.4.4 [36]. The pairwise p-distances and numbers of nucleotide differences between *Encotyllabe* species were calculated using MEGA v6 [37] (Table 2). Sequences of *Neobenedenia* sp. (Capsalidae) were used as outgroup (Table 1).

**Table 1.** GenBank sequences accession numbers for *Encotyllabe* spp., host species, family, locality, reference and the species used as outgroup phylogenetic analyses.

| Species | Host | Family | Locality | GenBank ID | | Reference |
|---|---|---|---|---|---|---|
| | | | | *cox*1 | LSU rRNA | |
| *E. bifurcatum* n. sp. | *Pagrus pagrus* | Sparidae | Brazil | | MT968928 | This study |
| *E. parvum* n. sp. | *Orthopristis ruber* | Haemulidae | Brazil | MT967362 | MT968927 | This study |
| *E.* cf. *spari* | *O. ruber* | Haemulidae | Brazil | | KY553149 | [4] |
| *E. antofagastensis* | *Anisotremus scapularis* | Haemulidae | Chile | JQ782836-40 | MT982166 | [3]/This study |
| *E. caballeroi* | *Gymnocranius audleyi* | Lethrinidae | Australia | | AF026112 | [38] |
| *E. caranxi* | *Pseudocaranx dentex* | Carangidae | Australia | | FJ971990 | [39] |
| *E. cheilodactyli* | *Cheilodactylus variegatus* | Cheilodactylidae | Chile | JQ782841-45 | MT982167 | [3]/This study |
| *E. chironemi* | *Chironemus marmoratus* | Chironemidae | Australia | | AF382054 | [40] |
| *Encotyllabe* sp.1 | *Pagellus bogaraveo* | Sparidae | Algeria | OL675214 | OL679678 | [12] |
| | | | | OL675215 | | [12] |
| | | | | OL675217 | | [12] |
| | | | | OL675220 | | [12] |
| | | | | OL675223 | | [12] |
| | | | | OL675214 | | [12] |
| *Encotyllabe* sp.2 | *Diplodus vulgaris* | Sparidae | Algeria | | OL679688 | [12] |
| | *Diplodus vulgaris* | Sparidae | Algeria | OL675225 | | [12] |
| | *Sparus aurata* | Sparidae | Algeria | OL675222 | | [12] |
| | | | | OL675226 | | [12] |
| *Neobenedenia* sp. | *Cheilodactylus variegatus* | Cheilodactylidae | Chile | JQ782846 | | [3] |
| *Neobenedenia* sp. | *Paralabrax humeralis* | Serranidae | Chile | | MK202450 | [41] |

**Table 2.** Pairwise sequence divergences for LSU rRNA (below diagonal, based on 780 bp) and *cox*1 mDNA (above diagonal, based on 269 bp) genes among species of *Encotyllabe*. *p*-distance and numbers of nucleotide differences are shown as percentage (%) with bp pairwise differences between parentheses. Due to the greater variability of the *cox*1 gene, the values are shown as a range.

| ID | | 1 | 2 | 3 | 4 | 5 | 6 | 7 | 8 | 9 |
|---|---|---|---|---|---|---|---|---|---|---|
| 1 | *Encotyllabe bifurcatum* (n = 2/0) | | | | | | | | | |
| 2 | *Encotyllabe parvum* (n = 3/4) | 0.8 (6) | | | 6.3–6.7 (17–18) | 8.6–8.9 (23–24) | | | 10.4–11.2 (28–30) | 11.2–11.5 (30–31) |
| 3 | *E.* cf. *spari* (n = 1/0) | 0.8 (6) | 0 | | | | | | | - |
| 4 | *E. antofagastensis* (n = 2/5) | 0.8 (6) | 0 | 0 | | 8.9–9.7 (25–26) | | | 10.0–11.5 (28–31) | 10.8–11.5 (29–31) |
| 5 | *E. cheilodactyli* (n = 2/5) | 1 (8) | 0.7 (5) | 0.7 (5) | 0.7 (5) | | | | 11.5–12.3 (31–32) | 11.5–12.3 (31–32) |
| 6 | *E. chironemi* (n = 1/0) | 0.7 (5) | 0.4 (3) | 0.4 (3) | 0.4 (3) | 0.5 (4) | | | | |
| 7 | *E. caballeroi* (n = 1/0) | 1.2 (9) | 1.2 (9) | 1.2 (9) | 1.2 (9) | 1.2 (9) | 1 (8) | | | |
| 8 | *Encotyllabe* sp.1 (n = 1/6) | 1 (8) | 1 (8) | 1 (8) | 1 (8) | 1.2 (9) | 0.8 (6) | 1.6 (12) | | 0.7–1.9 (2–4) |
| 9 | *Encotyllabe* sp.2 (n = 1/3) | 1 (8) | 1.2 (9) | 1.2 (9) | 1.2 (9) | 1.4 (11) | 1 (8) | 1.6 (12) | 0.3 (2) | |
| 10 | *E. caranxi* (n = 1/0) | 1.4 (5) | 1.4 (5) | 1.4 (5) | 1.4 (5) | 1.4 (5) | 1.4 (5) | 2 (7) | 1.4 (5) | 1.4 (5) |

Number of sequences for the LSU rRNA/*cox*1 gene indicated in parentheses.

## 3. Results

Class Monogenea van Beneden, 1858

Family Capsalidae Baird, 1853

*Encotyllabe bifurcatum* n. sp. Figure 1A–E

Type—host: *Pagrus pagrus* (Linnaeus) (Perciformes: Sparidae), red porgy.

Type—locality: Cabo Frio, Rio de Janeiro, Brazil (22°55′ S, 41°58′ W).

Type—material: Holotype (CHIOC 39975 a) and paratypes (CHIOC 39975 b-c-d) were submitted to the Helminthological Collection Instituto Oswaldo Cruz.

Site in host: Pharyngeal plates.

Prevalence and intensity: Prevalence: 69%; intensity: 1–12 per fish.

Representative DNA sequences: MT968928 (LSU rRNA)

Etymology: The specific name "*bifurcatum*" relates to the bifurcate MCO of this species.

*Description*

Measurements based on eleven mounted and stained adult worms: Body proper ellipsoidal 1780–2530 (2210, n = 9) long, 733–1120 (891, n = 9) wide, distinct ventral concavity; anterior attachment organs bearing two muscular suckers, 110–150 (127, n = 7) long, 134–184 (157, n = 7) wide, in each anterolateral region and surrounded by a flabellate lobe; haptor pedunculate, bell-shaped 415–588 (508, n = 10) in diameter, with thin marginal membrane 27–39 (35, n = 2) long, peduncle 556–678 (630, n = 9) long, 216–279 (238, n = 9) wide (Figure 1A); haptor armed with a pair of large anchors, 213–287 (258, n = 9) long, a pair of small anchors 29–32 (30, n = 9) long and 14 homogeneously distributed marginal hooks 10–15 (13, n = 9) long (Figure 1C–E); mouth surrounded by digitiform processes leading to a pharynx of irregular shape, 133–217 (152, n = 11) long and 156–243 (215, n = 11) wide; intestinal caeca branched are not confluent posteriorly; two pairs of eyespots at the level of pharynx; testes oval, juxtaposed, pre-equatorial, 246–380 (299, n = 11) long and 140–281 (211, n = 11) wide; Goto's glands are not observed; vas deferens sinistral winding anteriorly, entering at the base of the MCO and enlarging to form an internal seminal vesicle; prostatic duct joins ejaculatory duct and opens at the tip of the MCO; MCO muscular 240–344 (297, n = 6) long, 94–108 (102, n = 6) in wide, with a prolongation oriented to the anterior region of the body; genital pore ventral on the left side of the pharynx; ovary oval, pretesticular, immediately posterior to the vitelline reservoir 92–167 (130, n = 10) long, 130–232 (187, n = 10) wide, with an intraovarian seminal receptacle (Figure 1B); uterus extends anterolaterally along the posterior wall of MCO bulb; ootype was not observed, apparently hidden by Mehlis' gland, slender uterus opens at genital pore; vaginal pore ventral on central region of vitelline reservoir, and ducts were not observed; vitelline reservoir preovarian on the left side of the ovary; vitelline follicles are extensive laterally and median fields, from MCO to base of peduncle; eggs were not observed.

*Differential diagnosis*

*Encotyllabe bifurcatum* n. sp. resembles those species with pre-equatorial and juxtaposed testes and a larger peduncle, namely, *E. spari* and *E. antofagastensis*. The main differences between the new species and the above-mentioned species are the shape of the MCO with a projection near the genital pore in the new species not identified in any other species of the genus, whereas in *E. spari* the MCO is elongate, and *E. antofagastensis* being turns through in the posterior direction. In addition, the body size in *E. bifurcatum* being is smaller (1780–2530, mean 2210) than *E. spari* (3700 Holotype, range 1500–3000 for six paratypes) and *E. antofagastensis* (1520–3140, mean 2430). Moreover, the vitelline reservoir is definitively preovarian in *E. spari* and *E. antofagastensis* but anterolateral in the new species. Molecular analysis based on the LSU rRNA shows that *E. bifurcatum* is well discriminated between congeneric species with strong statistical support (ML = 92; BI = 0.99) (Table 2).

The type host for *E. spari* is *Sparus microcephalus* (Sparidae) from Japan, but it is also found in two non-related hosts in the Inner Sea (Japan): the haemulid *Plectorhynchus pictus* and the serranid *Epinephelus akaara*. The type host for *E. antofagastensis* is the haemulid *Anisotremus scapularis* from the southeastern Pacific. The new species is a parasite of a sparid (*Pagrus pagrus*) but from the coast of Brazil.

Examination of the specimen from *P. pagrus* identified as *E. spari* (CHIOC 34531) [23] showed that it belongs to the new species. Specimens found previously in the same host [25] also belong to the new species.

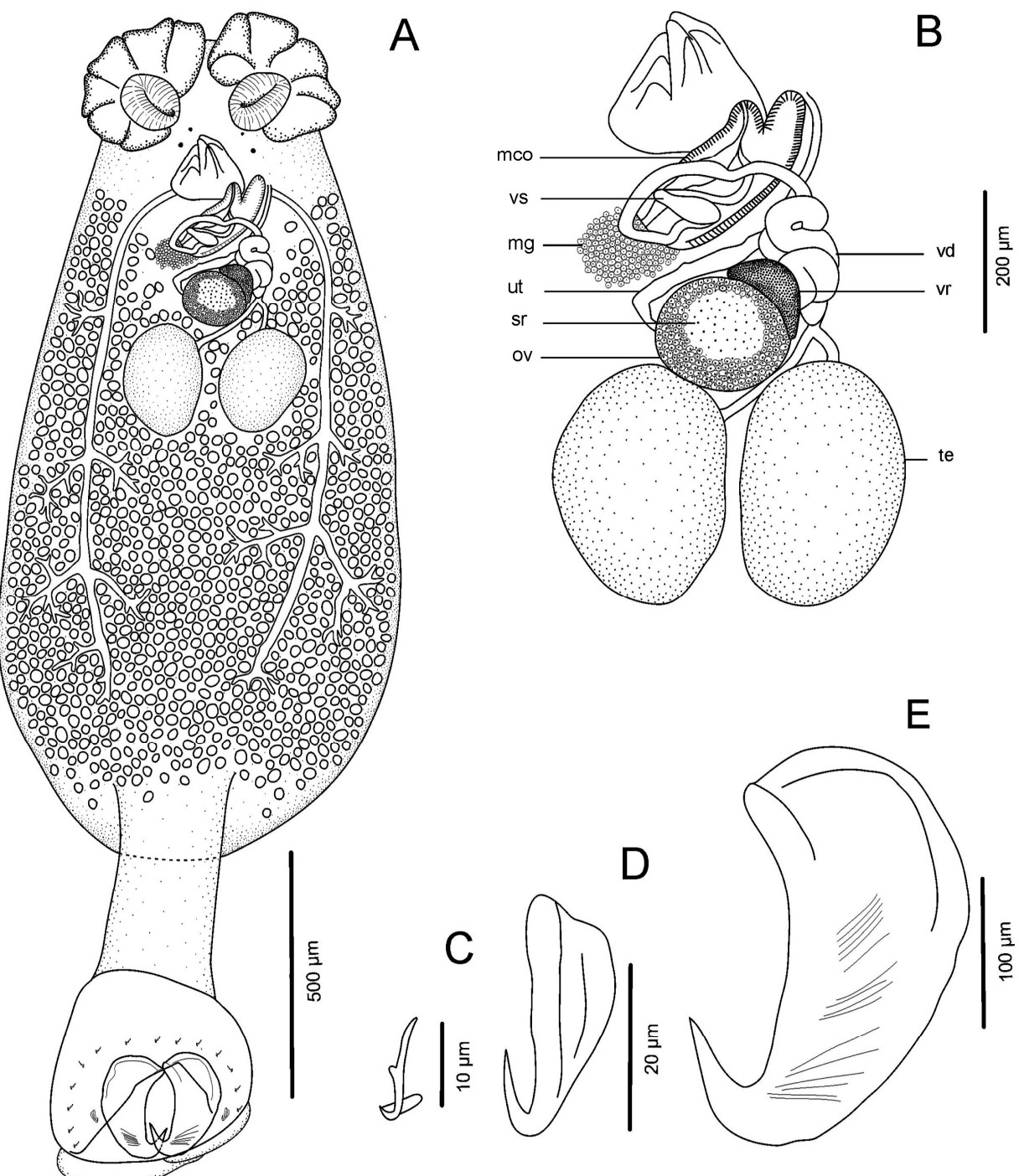

**Figure 1.** *Encotyllabe bifurcatum* n. sp. ex *Pagrus pagrus*; (**A**), Ventral view of the holotype; (**B**), Reproductive system; (**C**), Marginal hook; (**D**), Small anchor; (**E**), Large anchor. *Abbreviations*: mco, male copulatory organ; vs., seminal vesicle; mg, Mehlis' gland; ut, uterus; sr, seminal receptacle; ov, ovary; te, testes; vr, vitelline reservoir; vd, vas deferens.

*Encotyllabe parvum* n. sp.
Type—host: *Orthopristis ruber* (Cuvier) (Perciformes: Haemulidae), corocoro grunt.
Type—locality: Cabo Frio, Rio de Janeiro, Brazil (22°55′ S, 41°58′ W).

Type—material: Holotype (CHIOC 39976 a) and paratypes (CHIOC 39976 b-c-d) were submitted to the Helminthological Collection Instituto Oswaldo Cruz.

Site in host: Pharyngeal plates.

Prevalence and intensity: Prevalence: 100%; intensity: 3–13 per fish.

Representative DNA sequences: MT968927 (LSU rRNA), MT967362 (*cox*1).

Etymology: The species name comes from Latin "*parva*"—small—in reference to the small size of testes.

*Description*

Measurements are based on 13 stained and mounted adult worms: Body proper bell-shaped, tapered anteriorly and wide posteriorly 2740–3590 (2960, n = 11) long, 814–1650 (1200, n = 11) wide, ventral concavity; anterior attachment organs bearing two muscular suckers, 170–281 (229, n = 13) long, 175–284 (235, n = 13) wide, in each anterolateral region and surrounded by a flabellate lobe; haptor pedunculate, bell-shaped 476–784 (641, n = 13) in diameter, with a thin marginal membrane 56–76 (66, n = 5) long, peduncle 405–651 (542, n = 11) long, 310–444 (387, n = 11) wide (Figure 2A), haptor armed with a pair of large anchors 249–339 (291, n = 11) long, small anchors 25–33 (29, n = 8) long and 14 homogeneously distributed marginal hooks 10–13 (12, n = 7) (Figure 2D–F); mouth surrounded by digitiform processes leading to a globular pharynx 248–395 (323, n = 12) long, 242–448 (345, n = 12) wide; intestinal ceca branched are not confluent posteriorly; two pairs of eyespots are at the level of the pharynx; testes oval, juxtaposed, pre-equatorial, 132–181 (150, n = 10) long and 82–175 (116, n = 10) wide; Goto's glands are not observed; vas deferens sinistral winding anteriorly, entering at the base of the MCO, enlarging to form an internal seminal vesicle; prostatic duct joins ejaculatory duct and opens at the tip of the MCO; MCO muscular 185–228 (209, n = 7) long and 89–123 (108, n = 7) wide; genital pore ventral on the left posterior part of the pharynx; ovary oval, pretesticular, immediately posterior to the vitelline reservoir, 127–250 (183, n = 13) long and 176–248 (214, n = 13) wide, with an intraovarian seminal receptacle (Figure 2B); uterus extends anterolaterally along the posterior wall of the MCO; vaginal pore on left ventral side of the vitelline reservoir, and ducts are not observed; vitelline reservoir preovarian, sinistral; vitelline follicles extensive laterally and in median fields, from the pharynx to the base of peduncle; eggs pyramidal, with four long and twisted filaments (Figure 2C).

*Differential diagnosis*

Only two species of *Encotyllabe* have been described with testes smaller than the ovary, namely, *E. caranxi* and *E. embiotocae*. Of those, *E. caranxi* can be differentiated for the elongated body shape, being the longest species described in the genus (11.26 mm), with testes located in the anterior third of the proper body, whereas in *E. parvum*, the body is bell-shaped, and testes are located in the anterior part and not reaching the midlevel of the body. The relationship between large anchors and small anchors in *E. embiotocae* varies between 6.8:1 on average, whereas this value reaches 10:1 in the new species, with the smaller anchors in the new species being proportionally smaller. The genital pore in *E. embiotocae* is on the left side of the pharynx, whereas *E. parvum* has the genital pore posterior to the pharynx. In the original description of *E. embioticae* [42], four specimens were found with testes smaller than the ovary and another four specimens with testes larger than the ovary; this could imply that some of these specimens were immature, whereas in our specimens, eggs were observed. For *E. parvum*, all the specimens showed testes smaller than the ovary.

Specimens of *Encotyllabe* previously identified as *E. spari* from *Anisotremus virginicus*, *Conodon nobilis* and *Haemulon sciurus* deposited in CHIOC and identified as *E. spari* in fact belong to the new species. Specimens identified as *E. spari* from *Orthopristis ruber* [22] also belong to *E. parvum*. All these specimens analyzed can be differentiated from *E. spari* and resemble *E. parvum* by a bell-shaped body, i.e., tapered anteriorly and wider posteriorly, testes smaller than the ovary, testes pre-equatorial not reaching the midlevel of body proper and an elliptical MCO. *Encotyllabe* cf. *spari* [4] also described from *Orthopristis ruber* and caught off Rio de Janeiro resemble *E. parvum* by the features cited above, in addition to

similar measures. Molecular data using *cox*1 support the presence of this new species. (Table 2).

Unfortunately, most of the records of *E. spari* from Brazil come from non-taxonomic papers (i.e., check list, parasite community studies or parasites as biological tags), and morphological characterization are not given, except for just three articles [4,11,20].

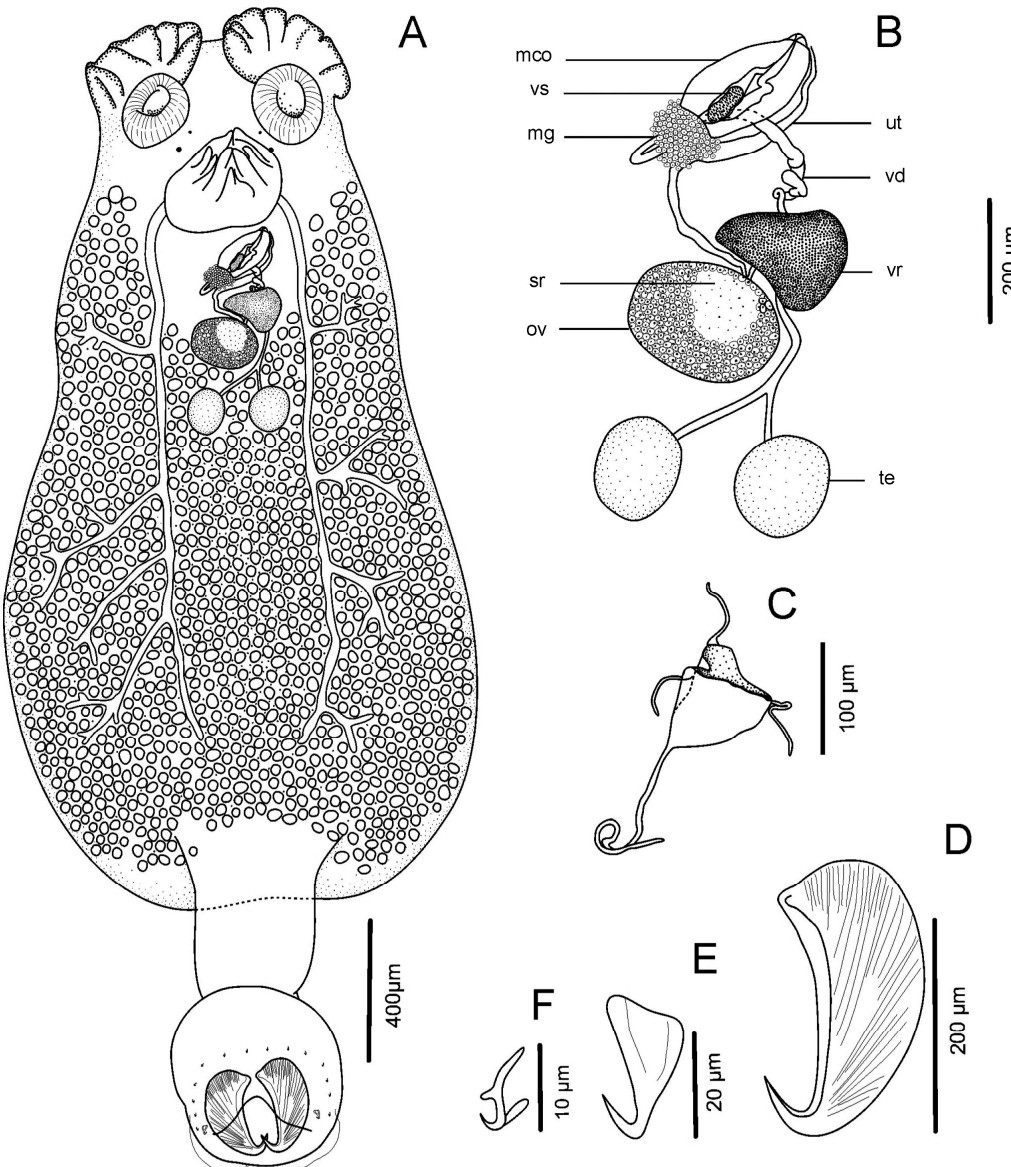

**Figure 2.** *Encotyllabe parvum* n. sp. ex *Orthopristis ruber*, (**A**), Ventral view of the holotype; (**B**), Reproductive system; (**C**), egg; (**D**), Large anchor; (**E**), Small anchor; (**F**), Marginal hook. *Abbreviations*: mco, male copulatory organ; vs, seminal vesicle; mg, Mehlis' gland; sr, seminal receptacle; ov, ovary; te, testes; vr, vitelline reservoir; vd, vas deferens; ut, uterus.

*Morphometric analysis*

Figure 3 presents the plot of specimens in a bidimensional space of the PCA. The first and second components explain 59.5% of the total variance. The first component, which explains 37.3% of the variance, was mainly associated with the proportional morphometric measurements of the ovary, attachment organs' average, testes' length, testes' width, large anchors' length and marginal hooks' length. The second component explains that 22.2% of the variance was associated with haptor diameter and body width.

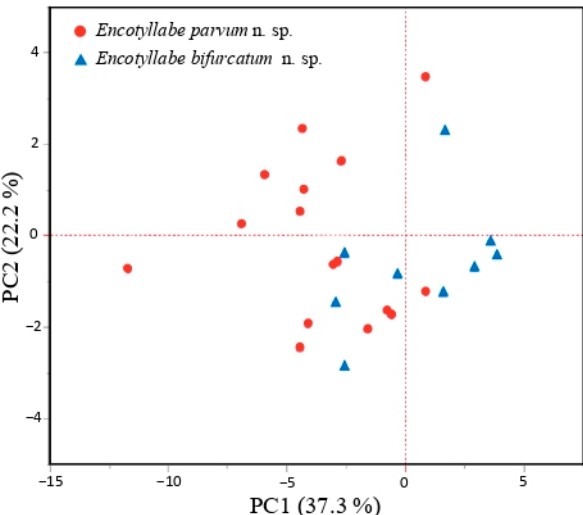

**Figure 3.** Principal components analysis based on proportional morphometric measurements standardized by total length from the new species of *Encotyllabe* from Brazil. Red circles: *E. parvum* ex *Orthopristis ruber*; Blue triangle *E. bifurcatum* n. sp. ex *Pagrus pagrus*.

*Molecular and phylogenetic analyses*

For the LSU rRNA, five sequences were obtained: three from *E. parvum* (852 bp) and two from *E. bifurcatum* (865 bp); the final alignment was 780 bp. The intraspecific genetic variability for both species was 0%; consequently, only one sequence was uploaded to Genbank for each new species (MT968927 and MT968928, respectively).

A phylogenetic tree using LSU rRNA gene based on ML and BI produced trees with similar topology (Figure 4). Each one of the new species was statistically supported in independent clades (ML > 90; BI > 0.88), except *E. parvum* n. sp. positioned in the same clade as *E.* cf. *spari* (KY553149) and *E. antofagastensis*, with 100% similarity between sequences (Table 2). The genetic distance between *E. parvum* and *E. bifurcatum* was 0.8% and differed at 6 nt positions. With the remaining congeneric species, *E. parvum* n. sp. was genetically distanced by 0.4% to 1.2% (Table 2). The genetic distance of *E. bifurcatum* n. sp. with congeneric species was 0.7% to 1.2% (five to nine nucleotides of difference).

Four sequences for *cox*1 were obtained from *E. parvum* (437 bp long) with 100% similarity. Despite several attempts, it was not possible to obtain sequences from *E. bifurcatum*. The final alignment of the *cox*1 gene data set was 269 bp. By contrast with LSU rRNA, *E. parvum* n. sp. constitutes an independent clade of *E. antofagastensis*, strongly supported by ML and BI (Figure 5). The genetic distance between *E. parvum* n. sp. and *E. antofagastensis* was on average 6.6% (18 nucleotides of difference). The genetic divergence with congeneric species is shown in Table 2.

Although molecular data for *E. spari* are not available in GenBank, there is a sequence (LSU rRNA gene KY553149) of *Encotyllabe* cf. *spari* from *Orthopristis ruber* caught in Brazil, included in our analyses.

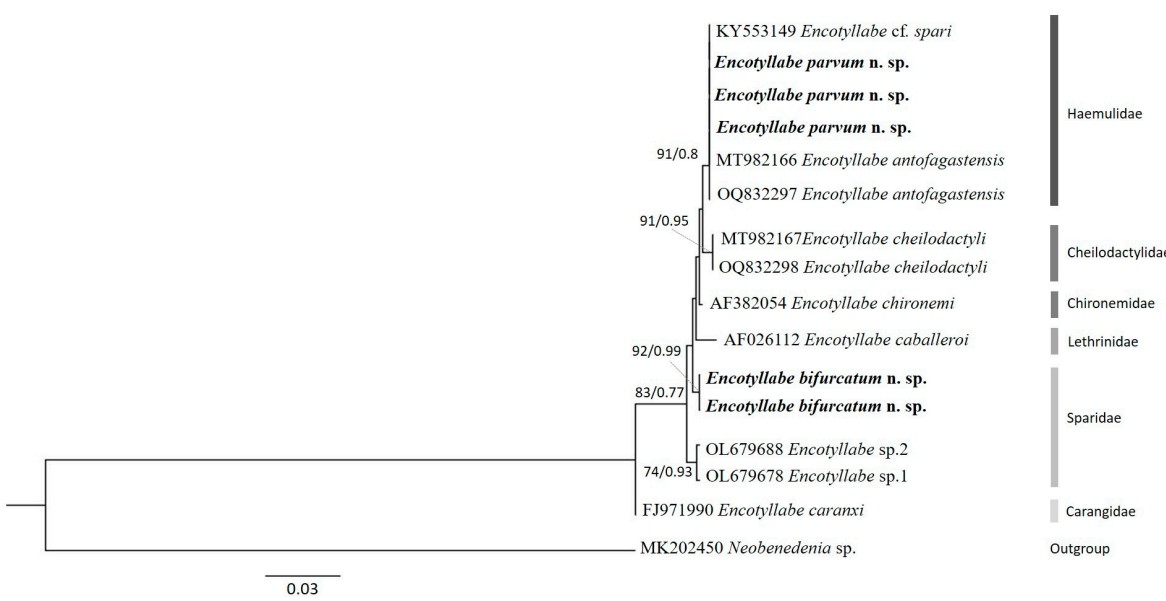

**Figure 4.** Phylogenetic tree based on LSU rRNA gene for *Encotyllabe* spp. inferred by maximum likelihood (ML) and Bayesian inference (BI). Numbers along branches indicate the bootstrap values obtained from the posterior probability support ML and BI (ML/BI).

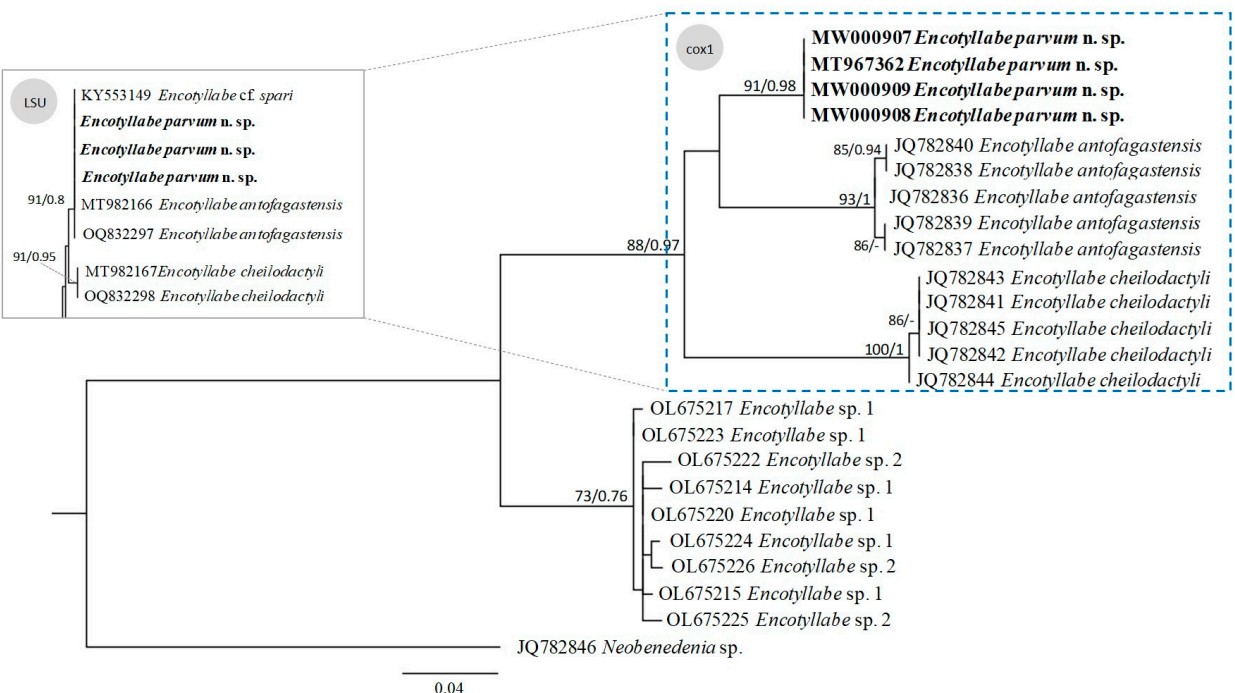

**Figure 5.** Phylogenetic tree based on cox1 mDNA gene for *Encotyllabe* spp. inferred by maximum likelihood (ML) and Bayesian inference (BI). Numbers along branches indicate the bootstrap values obtained from the posterior probability support ML and BI (ML/BI).

## 4. Discussion

Monogeneans are among the most host-specific parasites and may be the most host-specific of all fish parasites [7]. When broad host specificity, i.e., many hosts for the same species, of Monogenea is studied under molecular scrutiny, the broad specificity is questioned. For instance, the "cosmopolitan" capsalid *Neobenedenia melleni*, recorded from more than 100 host species in five orders, may be a complex of species, as suggested by molecular studies [43], and the related *Benedenia seriolae*, recorded as a parasite of natural

populations of *Seriola* spp. in Japan, Australia, and Chile, as well as in farmed conditions around the world, is also a species complex, as demonstrated by molecular evidence [44,45].

When the hosts for the recognized species of *Encotyllabe* were analyzed, an interesting picture became evident: five species (*E. antofagastensis*, *E. cheilodactyli*, *E. fotedari*, *E. lutjani*, and *E. xiamenensis*) have been recorded from one host species [3,13,46], and two species have been recorded from two different but related host species. *E. caranxi* was found in three species of *Caranx* and one of the related *Pseudocaranx* (Carangidae) [13], and *E. embiotocae* was found to be a parasite of *Cymatogaster aggregata* and *Amphistichus argenteus* (Embiotocidae) [42]. Exceptions are *E. caballeroi*, reported as a parasite of three species of *Lethrinus*, *Gymnocranius audleyi* (Lethrinidae), *Scolopis monogramma* and *Scolopis* sp. (Nemipteridae) from Australia, New Caledonia, the Philippines and Vietnam [13,47,48]; *E. kuwaitensis*, a parasite of *Caranx sexfasciatus*, *Caranx* sp. (Carangidae) and *Plectorhinchus schotaf* (Haemulidae) from the Arabian Gulf and the Mediterranean Sea [49,50]; and *E. souzalimae*, which has been reported from two nonrelated hosts from Brazil: *Trichiurus lepturus* (Trichiuridae) and *Thyrsitops lepidopoides* (Gempylidae) [5,20]. *E. pagrosomi* has been recorded from seven host species belonging to three families in the Galapagos Islands, Mexico (Pacific coast), Australia, Venezuela and Peru [13,51,52]. Finally, *E. spari* has been recorded from at least 25 host species in nine families and two orders in Japan, the Arabian Gulf, Brazil, Vietnam, Venezuela, Argentina and the Mediterranean Sea [4,5,13,16–25,49–51,53,54]. With regard to the Brazilian records, *E. spari* (but also *E.* cf. *spari*) has been recorded from at least four species of Haemulidae, *Conodon nobilis*, *Anisotremus surinamensis*, *Haemulon sciurus* and *Orthropristis ruber*; one species of Sparidae, *Pagrus pagrus* (Sparidae); two species of Sciaenidae (*Menticirrhus americanus*, *Micropogonias furnieri*); and one species of Dactylopteridae (*Dactylopterus volitans*) [4,16,17,20,21,24,25,55] (Table 3). Our data challenge the low host specificity of *Encotyllabe* from Brazil; worms from the sparid *Pagrus pagrus* and those from the haemulid *Orthopristis ruber* belong to different species, as demonstrated by molecular tools (Figure 3). Surprisingly, the two species from Haemulidae (*E. parvum* and *E. antofagastensis*) form a well-supported clade, suggesting additional evidence of high host specificity.

The taxonomic status of members of *Encotyllabe* is not easy to clarify, as traditional taxonomy is based on body shape, the relative sizes of several organs, the shape and relative position of the testes, the MCO shape, the extension of the vitellaria, and the size and shape of the anchors as well as the relative distances between different organs [3,5,14,38,56]. Additional characteristics, such as the relative position of the testes, have also been suggested, but multivariate analysis (i.e., principal component analysis) of proportions of different organs in relation to body size, rather the analysis or comparison of raw measurements, will be an adequate tool to discriminate species in this genus as suggested early for this genus [3] but also in *Acanthocotyle* [15]. Our results confirm the host specificity of *Encotyllabe* suggesting that previous records of *E. spari* in hosts other than *Pagrus pagrus* and *Orthopristis ruber* could represent new species (see Supplementary Material Table S2 for comparative measurements of the new species and those given for *E. spari*, and *E.* cf. *spari* as well the original description). This finding also applies to other species, particularly *E. spari*, *E. pagrosomi* and *E caballeroi*, with nonrelated hosts and from different geographical localities.

**Table 3.** *Encotyllabe* spp. accepted as valid in this study, recorded hosts, host family, locality and reference. Most of the references previous 2000, follow Egorova (2000) [13].

| Species | Recorded Hosts | Host Family | Locality | References |
|---|---|---|---|---|
| *E. antofagastensis* | *Anisotremus scapularis* | Haemulidae | Chile | [3] |
| *E. bifurcatum* | *Pagrus pagrus* | Sparidae | Brazil | This study |
| *E. caballeroi* | *Gymnocranius audleyi* | Lethrinidae | Australia | [13] |
| *E. caballeroi* | *Lethrinus miniatus* (*) | Lethrinidae | Australia; N. Caledonia | [13,17,18] |
| *E. caballeroi* | *Lethrinus nebulosus* | Lethrinidae | Philippines | [13] |
| *E. caballeroi* | *Scolopsis monogramma* | Nemipteridae | Australia | [13] |
| *E. caballeroi* | *Scolopsis* sp. | Nemipteridae | Vietnam | [13] |
| *E. caranxi* | *Caranx lutescens* | Carangidae | Australia | [13] |
| *E. caranxi* | *Caranx sexfaciatus* | Carangidae | Mediterranean Sea | [13] |
| *E. caranxi* | *Caranx* sp. | Carangidae | Australia | [13] |
| *E. caranxi* | *Pseudocaranx dentex* | Carangidae | Australia | [13] |
| *E. cheilodactylid* | *Cheilodactylus variegatus* | Cheilodactylidae | Chile | [3] |
| *E. embiotocae* | *Amphistichus argenteus* | Embiotocidae | California USA | [13] |
| *E. embiotocae* | *Cymatogaster aggergata* | Embiotocidae | California USA | [13] |
| *E. kuwaitensis* | *Caranx sexfasciatus* | Carangidae | Mediterranean Sea | [13] |
| *E. kuwaitensis* | *Caranx* sp. | Carangidae | Arabian Gulf | [13] |
| *E. kuwaitensis* | *Plectorhinchus shotaf* | Haemulidae | Arabian gulf | [18] |
| *E. lutjanid* | *Lutjanus johni* | Lutjanidae | India | [13] |
| *E. pagrosomi* | *Caulolatilus princeps* | Malacanthidae | Peru | [14] |
| *E. pagrosomi* | *Caulolatilus* sp. | Malacanthidae | Galapagos Islands | [13] |
| *E. pagrosomi* | *Chrysophrys auratus* | Sparidae | Australia | [13] |
| *E. pagrosomi* | *Haemulon steindachneri* | Haemulidae | Venezuela | [6] |
| *E. pagrosomi* | *Orthopristis ruber* | Haemulidae | Venezuela | [6] |
| *E. pagrosomi* | *Pagrosomus auratus* | Sparidae | Australia | [13] |
| *E. pagrosomi* | *Pomadasys macracanthus* | Haemulidae | Mexico | [13] |
| *E. parvum* | *Orthorpistis ruber* | Haemulidae | Brazil | This study |
| *E. souzalimae* | *Thyrsitops lepidopoides* | Gempylidae | Brazil | [11] |
| *E. souzalimae* | *Trichiurus lepturus* | Trichiuridae | Brazil | [5] |
| *E. spari* | *Acanthopagrus bifasciatus* | Sparidae | Arabian Gulf | [18] |
| *E. spari* | *Anisotremus surinamensis* | Haemulidae | Brazil | [35] |
| *E. spari* | *Argyrops spimniofer* | Sparidae | Arabian Gulf | [18] |
| *E. spari* | *Carangoides bajad* | Carangidae | Arabian Gulf | [18] |
| *E. spari* | *Conodon nobilis* | Haemulidae | Brazil | [35] |
| *E. spari* | *Epinephelus akaara* | Serranidae | Japan | [13] |
| *E. spari* | *Gymnocranius griseus* | Lethrinidae | Vietnam | [13] |
| *E. spari* | *Haemulon Sciurus* | Haemulidae | Brazil | [13] |
| *E. spari* | *Lethrinus nebulosus* | Lehtrinidae | Japan | [13] |
| *E. spari* | *Menticirrhus americanus* | Sciaenidae | Brazil | [7] |
| *E. spari* | *Micropogonias furnieri* | Sciaenidae | Brazil | [35] |
| *E. spari* | *Nemipterus virgatus* | Nemipteridae | Japan | [13] |
| *E. spari* | *Orthopristis ruber* | Haemulidae | Brazil | [9] |
| *E. spari* | *Pagrus major* | Sparidae | Japan | [13] |
| *E. spari* | *Pagrus pagrus* | Sparidae | Brazil | [23] |
| *E. spari* | *Parapristipoma trilineatus* | Haemulidae | Japan | [13] |
| *E. spari* | *Plectorhinchus cinctus* | Haemulidae | Arabian Gulf | [19] |
| *E. spari* | *Plectorhinchus pictus* | Haemulidae | Arabian Gulf | [13,19] |
| *E. spari* | *Plectorhinchus schotaf* | Haemulidae | Arabian Gulf | [19] |
| *E. spari* | *Plectorhinchus* sp. | Haemulidae | Vietnam | [13] |
| *E. spari* | *Plectorhinchus* spp. | Haemulidae | Arabian Gulf | [18] |
| *E. spari* | *Sebastes inermis* | Sebastidae | Japan | [13] |
| *E. spari* | *Sparus aurata* | Sparidae | Mediterranean Sea | [27] |
| *E. spari* | *Sparus macrocephalus* | Sparidae | Japan | [13] |
| *E. spari* | *Umbrina canosai* | Sciaenidae | Argentina | [4] |
| *E. spari* | *Upeneus tragula* | Mullidae | Japan | [13] |
| *E. xiamenesis* | *Pagrosomus major* | Sparidae | Taiwan | [23] |

**Supplementary Materials:** The following supporting information can be downloaded at: https://www.mdpi.com/article/10.3390/d15060706/s1, Table S1. Morphometry of the recognized species of *Encotyllabe* according to original description or redescription. Table S2. Comparative data for *Encotyllabe spari* registered in Brazil, the original description, *Encotyllabe bifurcatum* and *Encotyllabe parvum* described in present study. Ref. [57] is cited in Supplementary Materials.

**Author Contributions:** N.T., J.L.L. and M.E.O. conceived and designed the study; N.T. and J.L.L. carried out the field work; F.A.S. performed molecular analyses. Additional analyses were performed by R.E. and M.E.O.; N.T., J.L.L. and M.E.O. wrote the manuscript. All authors have read and agreed to the published version of the manuscript.

**Funding:** FAS was supported by "Programa Semilleros de Investigación" DGI-Universidad de Antofagasta-grant 5303. Additional support was provided by the Millenium Instituto of Oceanography (IMO), ANID-ICN12_019N to RE and Plan de Fortalecimiento Universidades Estatales-Chile RES21992 (MEO).

**Institutional Review Board Statement:** This study did not consider experiments with live animals. All fishes were obtained from commercial catches, and none of the species are subject to conservation measures.

**Informed Consent Statement:** Not applicable.

**Data Availability Statement:** Data are available as Supplementary Materials.

**Acknowledgments:** We thank Tomáš Scholz (Institute of Parasitology, České Budějovice) for helpful suggestions to an early draft of the manuscript. NLT and JLL were supported by CNPq (Conselho Nacional de Desenvolvimento Científico e Tecnológico, Brazil).

**Conflicts of Interest:** The authors declare no conflict of interest.

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
