# Peer review of "Two New Species of Encotyllabe (Monogenea: Capsalidae) from Brazil: Morphological and Molecular Evidence"

_diversity, doi:10.3390/d15060706_

Round 1
Reviewer 1 Report
This manuscript presents descriptions of two new species of Enctoyllabe, based on morphological and molecular evidence, from fish in Brazil. I applaud the authors for such a comprehensive analysis, including a variety of techniques to ensure that their species are well described and characterised (unlike the majority of descriptions of species within the genus).
I found the manuscript to be well written with the arguments and discussion well presented and thought out. However, there are numerous small grammatical and spelling errors and I would recommend the authors have a native English-speaker read over the manuscript. English is a hard language, even for native speakers, to try and explain why certain things happen in certain locations and not others – the fact that this manuscript is so well written is a credit to the authors and I mean no disrespect.
There are a number of comments below that can be addressed:
Line 20: change “;” to “,”
Line 29: insert “previously” after “specimens”
Line 41 (and elsewhere): “inquirenda” and/or “species inquirenda” need to be in italics
Line 45-49: change “;” to “,”
Line 64: remove “posteriorly”
Line 73: insert “,” following “described”
Line 77: should this be “peduncle”?
Line 79, 81-86, 368, 472, 488: italicise genus and species names
Table 1: “Species”
Table 1: the formatting of this Table is not great, with many of the names going over two lines. Can the Table be formatted in landscape or with wider margins as in Table 2?
Line 231: spelling of Encotyllabe
Line 265: italicise “parva”
Line 306: I would suggest adding a line that eggs were found in your specimens, indicating that they were reproductively mature.
Line 324: The labels are incorrect for D-F
Line 346: delete “t” following the bracket
Line 365: spelling “caught”
Line 366: spelling “analyses”, and need a stop at the end of the sentence
Figure 5: where does the third sequence of E. antofagastensis belong?
Line 399, 404: Sea
Line 423, 526: delete the underline
Line 433: remove stop after MEO
Line 456: Remove stop at beginning of line
Line 490, 496, 499: check format of reference
Line 601: ?
See comments above - some editing is required
Author Response
Reviewer 2
Open Review
(x) I would not like to sign my review report
( ) I would like to sign my review report
Quality of English Language
(x) I am not qualified to assess the quality of English in this paper
( ) English very difficult to understand/incomprehensible
( ) Extensive editing of English language required
( ) Moderate editing of English language
( ) Minor editing of English language required
( ) English language fine. No issues detected
|
Yes |
Can be improved |
Must be improved |
Not applicable |
|
|
Does the introduction provide sufficient background and include all relevant references? |
(x) |
( ) |
( ) |
( ) |
|
Are all the cited references relevant to the research? |
(x) |
( ) |
( ) |
( ) |
|
Is the research design appropriate? |
(x) |
( ) |
( ) |
( ) |
|
Are the methods adequately described? |
(x) |
( ) |
( ) |
( ) |
|
Are the results clearly presented? |
(x) |
( ) |
( ) |
( ) |
|
Are the conclusions supported by the results? |
(x) |
( ) |
( ) |
( ) |
Comments and Suggestions for Authors
Line 79 – Encotyllabe in italics
Line 81-86 – same issue, be careful writing scientific names, should be in italics. A problem using the template
Line 97- aperiodically? Is that correct? Deleted
Table 1 – Specie
Specie and species are both nouns, meaning a person, place, or thing. “Species” are living beings, while “specie” is a thing. changed
Line 231 – Correct the EDncotyllabe changed
Line 346- extra t deleted
Line 348- species changed
Line 366 – analyses not snalyses
Discussion section can be improved.
I missed a discussion about the differences in the genes results, why LSU did not show the genetic divergence among species? And COX1 did?
Please note that is well known that LSU rRNA sequences are widely used for working out evolutionary relationships (phylogeny) among organisms, since they are conserved and of ancient origin, whereas cox1 gene shows a high mutational rate and is mainly used in barcoding, that is species identification.
Also, it will be very interesting to get some other samples of the very common species from other localities. Please, discuss the possibility of improvements to the study in the Discussion section.
It is not easy to get samples from other localities. In the last lines of discussion we comment that our findings (based on an integrative taxonomic approach) could clarify the status of some species with "broad" host specificity
Submission Date
21 April 2023
Date of this review
04 May 2023 14:56:52
Reviewer 2 Report
Line 79 – Encotyllabe in italics
Line 81-86 – same issue, be careful writing scientific names, should be in italics.
Line 97- aperiodically? Is that correct?
Table 1 – Specie
Specie and species are both nouns, meaning a person, place, or thing. “Species” are living beings, while “specie” is a thing.
Line 231 – Correct the EDncotyllabe
Line 346- extra t
Line 348- species
Line 366 – analyses not snalyses
Discussion section can be improved.
I missed a discussion about the differences in the genes results, why LSU did not show the genetic divergence among species? And COX1 did?
Also, it will be very interesting to get some other samples of the very common species from other localities. Please, discuss the possibility of improvements to the study in the Discussion section
Author Response
Reviewer 2
Open Review
(x) I would not like to sign my review report
( ) I would like to sign my review report
Quality of English Language
(x) I am not qualified to assess the quality of English in this paper
( ) English very difficult to understand/incomprehensible
( ) Extensive editing of English language required
( ) Moderate editing of English language
( ) Minor editing of English language required
( ) English language fine. No issues detected
|
Yes |
Can be improved |
Must be improved |
Not applicable |
|
|
Does the introduction provide sufficient background and include all relevant references? |
(x) |
( ) |
( ) |
( ) |
|
Are all the cited references relevant to the research? |
(x) |
( ) |
( ) |
( ) |
|
Is the research design appropriate? |
(x) |
( ) |
( ) |
( ) |
|
Are the methods adequately described? |
(x) |
( ) |
( ) |
( ) |
|
Are the results clearly presented? |
(x) |
( ) |
( ) |
( ) |
|
Are the conclusions supported by the results? |
(x) |
( ) |
( ) |
( ) |
Comments and Suggestions for Authors
Line 79 – Encotyllabe in italics
Line 81-86 – same issue, be careful writing scientific names, should be in italics. A problem using the template
Line 97- aperiodically? Is that correct? Deleted
Table 1 – Specie
Specie and species are both nouns, meaning a person, place, or thing. “Species” are living beings, while “specie” is a thing. changed
Line 231 – Correct the EDncotyllabe changed
Line 346- extra t deleted
Line 348- species changed
Line 366 – analyses not snalyses
Discussion section can be improved.
I missed a discussion about the differences in the genes results, why LSU did not show the genetic divergence among species? And COX1 did?
Please note that is well known that LSU rRNA sequences are widely used for working out evolutionary relationships (phylogeny) among organisms, since they are conserved and of ancient origin, whereas cox1 gene shows a high mutational rate and is mainly used in barcoding, that is species identification.
Also, it will be very interesting to get some other samples of the very common species from other localities. Please, discuss the possibility of improvements to the study in the Discussion section.
It is not easy to get samples from other localities. In the last lines of discussion we comment that our findings (based on an integrative taxonomic approach) could clarify the status of some species with "broad" host specificity
S